# Determination of Drying Patterns of Radish Slabs under Different Drying Methods Using Hyperspectral Imaging Coupled with Multivariate Analysis

**DOI:** 10.3390/foods9040484

**Published:** 2020-04-12

**Authors:** Dongyoung Lee, Santosh Lohumi, Byoung-Kwan Cho, Seung Hyun Lee, Hyunmo Jung

**Affiliations:** 1Department of Molecular Biosciences and Bioengineering, University of Hawaii, Honolulu, HI 96822, USA; lee272@hawaii.edu; 2Department of Biosystems Machinery Engineering, Chungnam National University, 99 Daehak-ro, Yuseong-gu, Daejeon 34134, Korea; santosh123@cnu.ac.kr (S.L.); chobk@cnu.ac.kr (B.-K.C.); 3Department of Smart Agriculture Systems, Chungnam National University, 99 Daehak-ro, Yuseong-gu, Daejeon 34134, Korea; 4Department of Logistic Packaging, Kyongbuk Science College, 634 Jisan-ro, Gisan-myeon, Chilgok-gun, Gyeongsangbuk-do 39913, Korea

**Keywords:** radish, moisture content, drying pattern, multivariate analysis, hyperspectral imaging

## Abstract

Drying kinetics and the moisture distribution map of radish slabs under different drying methods (hot-air drying (HAD), microwave drying (MD), and hot-air and microwave combination drying (HMCD)) were determined and visualized by hyperspectral image (HSI) processing coupled with a partial least square regression (PLSR)-variable importance in projection (VIP) model, respectively. Page model was the most suitable in describing the experimental moisture loss data of radish slabs regardless of the drying method. Dielectric properties (DP, ε) of radish slices decreased with the decrease in moisture content (MC) during MD, and the penetration depth of microwaves in radish was between 0.81 and 1.15 cm. The PLSR-VIP model developed with 38 optimal variables could result in the high prediction accuracies for both the calibration ($R_{cal}^{2}=0.967$ and RMSEC=4.32%) and validation ($R_{val}^{2}=0.962$ and RMSEC=4.45%). In visualized drying patterns, the radish slabs dried by HAD had a higher moisture content at the center than at the edges; however, the samples dried by MD contained higher moisture content at the edges. The nearly uniform drying pattern of radish slabs under HMCD was observed in hyperspectral images. Drying uniformity of radish slabs could be improved by the combination drying method, which significantly reduces drying time.

## 1. Introduction

Radish is a popular root vegetable with high moisture content and good effect on digestion [1] It is processed into a variety of products such as dried, salted, pickled, and fermented products in Asia [2]. The use of drying for radish is accepted as a simple method to enhance stability and extend shelf life. Conventional convective drying or sun drying is commonly employed to dehydrate radish; however, it requires quite a long drying time and often causes quality deterioration, such as high shrinkage, surface cracking, and low rehydration rate. Microwave drying (MD) as an alternative to hot-air drying (HAD) significantly improves the drying rate while maintaining the quality of dried food materials or agricultural products. The interaction between electromagnetic energy from microwave and agricultural products should be investigated when using microwaves [3]. Dielectric properties (DP) ($\varepsilon =\varepsilon^{\prime }-j\varepsilon^{\prime \prime }$), which consist of dielectric constant ($\varepsilon^{\prime }$) and dielectric loss factor ($\varepsilon^{\prime \prime }$), should be determined to understand this interaction. In addition, the properties are significantly changed depending on the moisture content (MC) and the temperature of the sample. Therefore, it is necessary to measure the DP of agricultural products in accordance with the change in MC during MD. Studies on the measurement of DP of agricultural products depending on temperature have been performed by several researchers; however, few studies have been conducted to measure the DP of agricultural products depending on MC. The main disadvantage of MD is non-uniform heating. In order to overcome the inherent problems of MD and HAD, a number of researchers have investigated the effect of hot-air and microwave combination drying (HMCD), and have reported that this combination drying method leads to a more effective drying process than individual MD or HAD methods [4]. Since the change in MC of agricultural products or food is a key indicator of the drying process, it is important to measure MC of agricultural products during drying, and the visualization of the MC distribution in agricultural products is necessary to ensure a uniform drying. Near infrared (NIR) spectroscopy and hyperspectral imaging (HSI) techniques were widely employed to determine MC of agro-food materials during drying because these techniques provide various information (i.e., chemical property, physicochemical property) of scanned agro-food materials.

NIR spectroscopy has been used as a non-destructive technique for prediction of MC in a wide range of food products including grains [5], fruits and vegetables [6], meat, and meat products [7]. The information of NIR spectra mainly derived from the O–H, C–H, and N–H bonds. Furthermore, the O–H bond, which is the only existing bond in water, has several characteristic absorption peaks in the NIR region (between 780 and 2500 nm) of the electromagnetic spectrum [6]. Hence, the MC of agro-food samples can be effectively detected by using NIR spectroscopy. Despite being effective for MC detection, spectroscopy-based techniques can only provide the spectral information from a small area of the sample. Furthermore, this technique cannot provide the spatial distribution and component visualization information. This information is very crucial when it is necessary to scan and visualize the entire sample based on its chemical components to acquire the physicochemical properties of the sample.

HSI technique, which utilizes an imaging component along with spectroscopy, is a powerful technique that simultaneously provides spectral and spatial information from the scanned sample. HSI has received considerable attention in the agro-food sector for quality analysis of a range of food materials [8,9,10,11]. Notably, HSI with image processing technique has a great potential for spatial visualization of the distribution of MC within agricultural products. Recently, Amjad et al. [12] determined the change in MC of potato slices during HAD process using HSI techniques. In the final stage of drying, uniform moisture distribution within the potato slice was observed in the moisture mapping results. Pu and Sun [13] evaluated the potential of hyperspectral imaging for visualization of the MC distribution in mango slices dried by hot-air and microwave-vacuum drying. A non-uniform moisture distribution was found in mango slices dried by microwave-vacuum drying. The MC in the center of the slices dried by microwave-vacuum drying was lower than in the four corners of the slices, which differs from the results obtained with HAD. In a similar study conducted by Liu et al. [14], moisture distribution maps of beef slices dried using microwaves were visualized using HSI technique. The resulting images indicated that the MC at the center of the meat slices was higher than at the edges. The aforementioned studies report different findings because of the change in the sample properties such as shape, size, and DP during the drying process [15,16]. It is necessary to confirm the drying uniformity of agricultural products, and the changes in DP of agricultural products depending on their MCs should be determined during MD.

Therefore, this study aimed to (i) explore the drying characteristics of radish slabs under different drying methods (HAD, MD, and HMCD methods), (ii) determine the changes in DP of radish depending on the MC, (iii) predict and visualize MC distribution of radish dried by different drying methods using a multivariate data analysis technique and an image processing strategy, (iv) improve the prediction efficiency of the multivariate model and accurate visualization of the moisture distribution map by selecting optimal variables representing the changes in sample MC, and (v) compare the obtained results with previously mentioned studies and suggest competent techniques for the uniform drying of radish samples.

## 2. Materials and Methods

### 2.1. Sample Preparation

Fresh Asian white radishes (*Raphanus sativus* L.) were purchased from a local agricultural market (Noeun wholesale market, Daejeon, Korea). The radishes were washed with tap water and then cut into slab shapes (40 ± 0.4 mm^2^ in area and 5 ± 0.2 mm in thickness) using a custom designed adjustable thickness cutter. A total of four radish slabs with an average weight of 32.23 ± 1.30 g were used for each drying experiment. The initial MC of radish sample was determined by convective drying at 105 °C for 24 h. The mean initial MC was 18.11 ± 2.61 kg water/kg dry matter.

### 2.2. Drying Equipment

Drying experiments were performed in the lab-scale microwave and hot-air combination dryer consisting of a microwave power unit and hot-air generator as shown in Figure 1. A microwave cavity of a domestic microwave oven (Mwx304sl, Whirlpool Co, Benton Harbor, MI, USA) was used as the drying chamber and the other parts of the oven were removed. In order to remove the evaporated water from the radish samples during drying, a small fan was mounted on the back side of the drying chamber. The microwave power unit could provide a power intensity of 0–1000 W at operating frequency of 2450 MHz. Depending on the changes in the environment of MD chamber, microwave intensity could be manually modified by using the phase control. The hot-air generator consisted of an electric heating coil, a centrifugal fan, and a proportional-integrative-derivative (PID) controller for air temperature and velocity. Hot-air temperature was measured using a K-type thermocouple (KK-K-30, Omega Engineering Inc., Stamford, CT, USA) connected to the PID controller. The generated hot air was supplied to the left side of the drying chamber through a circular duct. The inlet air velocity measured by a hygro-anemometer (HHC261, OMEGA Engineering, Inc., Stamford, CT, USA) was 5 m/s.

### 2.3. Drying Procedure

The different drying characteristics of radish slabs dried using different drying methods (HAD, MD, and HMCD) were evaluated in this study. Four radish slabs were spread over on a circular mesh tray made of Teflon thread. A total of 112 radish slab samples (HAD: 36, MVD: 40, CD: 36) were used.

#### 2.3.1. HAD Procedure

The applied hot-air temperature and velocity were 70 °C and 5 m/s, respectively. The weight of the radish slabs during the drying process was measured by a digital electronic balance (FX-3000i, A&D Company Ltd., Tokyo, Japan) at 15 min interval from 0 to 135 min.

#### 2.3.2. MD Procedure

As a preliminary experiment, radish slabs were dried by MD with different microwave power intensities (90–900 W at increments of 90 W); however, charring was observed at the end of the drying process for most the conditions, except for a 90 W power intensity. Although 90 W was a suitable power intensity for drying radish slabs, a relatively lower microwave power (50 W) was applied to prevent excessive shrinkage of the dried radish slabs. The weight of radish slabs was measured in 5 min intervals and the total drying time was 50 min.

#### 2.3.3. HMCD Procedure

HMCD was conducted by simultaneously applying a continuous microwave power of 50 W and a hot-air temperature of 70 °C at a velocity of 5 m/s. The weight of the sample was measured in intervals of 5 min and HNCD was completed after 45 min.

### 2.4. Drying Kinetics

The moisture ratio of the radish slabs samples during each different drying experiment was calculated using the following Equation (1):(1)$$\mathrm{Moisture}\;\mathrm{ratio}\;\left(\mathrm{MR}\right)=\frac{M-M_{e}}{M_{0}-M_{e}}$$
where M, $M_{0}$, and $M_{e}$ are the moisture content at any time, initial moisture content, and equilibrium moisture content, respectively.

The drying kinetics models (six semi-theoretical models and one empirical model as summarized Table 1 were investigated to analyze the drying behavior of radish slabs under different drying methods. Model parameters were estimated by nonlinear regression analysis using SPSS 24.0 software (IBM SPSS Statistics, Chicago, IL, USA). The goodness of fit of the experimental data for the different models was evaluated by establishing the determination coefficient ($R^{2}$), root mean square error (RMSE), and chi-square ($\chi^{2}$). RMSE and $\chi^{2}$ were determined by Equations (2) and (3), respectively.
(2)$$\mathrm{RMSE}\;={\left[\frac{1}{N}\overset{N}{\underset{i=1}{\sum }}{\left(MR_{pre,i}-MR_{exp,i}\right)}^{2}\right]}^{\frac{1}{2}}$$
(3)$$\chi^{2}=\frac{\sum_{i=1}^{N}{\left(MR_{pre,i}-MR_{exp,i}\right)}^{2}}{N-z}$$
where *MR_exp,i_* is the *i*^th^ experimental moisture ratio, *MR_pre,i_* is the *i*^th^ predicted moisture ratio, *N* is the number of observations, and *z* is the number of drying constants.

### 2.5. Measurement Procedure of Dielectric Properties

Open-ended coaxial-probe measurement system consisting of a dielectric probe kit (85070E (Performance probe), Keysight Technologies Co., Santa Rosa, CA, USA) and a vector network analyzer (VNA) (N9923A, Keysight Technologies Co, Santa Rosa, CA, USA) was employed to determine the DP of radish. Since this measurement method requires the close contact between the probe and the sample, the performance dielectric probe, which is suitable for most demanding applications and useful for measuring DP of small samples, was employed for DP measurement. The diameter (d) of the nickel-plated tungsten center conductor in the performance probe was 1.6 mm. The minimum sample thickness required for DP measurement with the performance probe is 3.2 mm (≈2d) regardless of the sample area to ensure the “infinite-medium” requirement [24,25]. During the DP measurement with the performance probe, the reflection coefficient (S_11_) of radish was measured with the VNA and then was converted to dielectric constant ($\varepsilon^{\prime }$) and dielectric loss factor ($\varepsilon^{\prime \prime }$) using 85070E software (Keysight Technologies Co, Santa Rosa, CA, USA). DP of radish was measured in the frequency range of 500–3000 MHz at 101 different frequencies points and DP at 2450 MHz was separately extracted from the entire DP data. During the DP measurement of radish, the VNA was turned on for at least 1 h before calibration. The calibration process was conducted using air, shorting block, and water stages.

The changes in the DP of radish during MD were measured using a similar experimental protocol of MD experiment. For the measurement of the DP of radish, the middle part of a radish was cut into slices (thickness: around 6 mm, diameter: around 100 mm) using a vegetable slicer. A radish slice was put on a circular mesh tray and MD was applied. The average weight and initial MC of the radish slices were 43.98 ± 0.79 g and 93.34% ± 0.65% (MC_w.b._), respectively. Three microwave power levels (810, 270, and 180 W) were applied to determine the effect of microwave power and MC on the change in the DP of radish during MD. The slice samples were taken out from the microwave chamber to measure the weight and DP values during MD, and the measurement time intervals for 810, 270, and 180 W were 30 s, 1 min, and 2 min, respectively. The DP of radish samples were measured at the center of the radish slice. In order to prevent the occurrence of error in DP measurement due to partially generated moisture on the surface of the slice sample during MD, the moisture on the surface was removed using paper wipes before DP measurement. As mentioned earlier, the radish slab with small thickness under MD with a power intensity over 90 W was charred at the end of drying stage; therefore, when the MC of radish cube samples reached approximately 20% (MC_w.b._), MD was stopped.

Based on the DP of radish slices with different MCs, the penetration depth of microwave was calculated from the following Equation (4):(4)$$d_{p}=\frac{C}{2\pi f\sqrt{2\varepsilon^{\prime }\left[\sqrt{1+{\left(\frac{\varepsilon^{\prime \prime }}{\varepsilon^{\prime }}\right)}^{2}}-1\right]}}$$
where $d_{p}$ is the penetration depth (m), *f* is the frequency (Hz), *c* is the speed of light in the free space (2.9989 × 10^8^ m/s), and $\varepsilon^{\prime }$ and $\varepsilon^{\prime \prime }$ are the measured values of dielectric constant and dielectric loss factor of radish, respectively.

### 2.6. Hyperspectral Image Instrumentation and Image Acquisition

A laboratory-based push-broom NIR-HSI system as shown in Figure 2 was applied to acquire the hyperspectral images of the radish slab samples. The system was composed of an illumination unit, a sensing module, and a conveyor unit. A total of six 100 W tungsten-halogen lamps (Light Bank, Ushio Inc., Japan) with fiber optics (three on each side) were utilized to illuminate the samples and the uniformity of illumination at each point of the illumination line was secured by measuring the Teflon sheet. In order to form hyperspectral images, the sensing module consisted of an objective lens with 25 mm focal length (Navitar, SWIR-25, Rochester, NY, USA), short-wave infrared (SWIR) camera (SWIR, Headwall Photonics, MA, USA), and an imaging spectrograph which disperses the collected light onto the SWIR camera. The SWIR camera covers the spectral range of 894–2504 nm with a spectral interval of approximately 5.85 nm hence a total of 275 variables in spectral dimension. The sensing unit was linked to the computer through frame grabber with a standard camera link cable. A translation stage was used for sample movement operated by a stepper motor. Commercial software provided by the camera manufacturer was used to synchronize the translation stage with the imaging camera, and for HSI data acquisition.

The radish slab samples were placed on the sample holder located on the translation stage that was perpendicularly 32 cm away from the camera lens, and then sample holder was moved to the camera field of view (FOV) for scanning radish samples line-by-line. The HSI data were collected with a 50 ms exposure time and the speed of translation stage was set to 4.55 mm/s to cover the spatial shape of the samples. Spectral and spatial data were obtained when there was movement in the camera’s FOV. Hyperspectral images of scanned samples were saved in raw format as 3D hypercube which consist of two spatial dimensions and one spectral dimension.

### 2.7. Image Correction and Spectral Data Extraction

After performing the hyperspectral scanning of the samples, white reference (X*ref*) and dark current (*X_dark_*) images were acquired to correct the HSI images. The white reference image was obtained using a white Teflon tile with approximately 99% reflectance, and by covering the lens with an opaque cap and turning off the light source, the dark current image (approximately 0% reflectance) was obtained. In order to extract the actual infrared (IR) response of the sample, the influence from both the white reference and the dark current images was removed by applying the following Equation (5):(5)$$X_{corrected\;}=\frac{\left(X_{raw})-(X_{dark}\right)}{\left(X_{ref})-(X_{dark}\right)}$$
where $X_{corrected}$ is the corrected hypercube and $X_{raw}$ is the acquired hyperspectral images of samples.

The corrected hypercube contained both the sample and the background (sample plate) pixels. Therefore, based on the pixel intensity, a threshold value was applied to the hyperspectral band image (1100 nm) to remove the irrelevant background pixels. Thus, a region of interest (ROI) step was performed on the thresholded hypercube to select only the pixels from the samples. The ROI for each sample was visually identified and pixels were manually selected. The schematic diagram of the experiment and hyperspectral imaging process is presented in Figure 3.

### 2.8. Reference Measurements

The initial MC of radish slab samples was 94.70% ± 0.73% in MC_w.b_. It was used as moisture reference value. The MC values of the radish slab samples during the different drying methods were measured and then the images of dried radish slab samples were obtained using the HSI system.

### 2.9. Data Processing and Multivariate Calibration Model

The main data processing used in this study is illustrated in Figure 3. The average spectra of each sample extracted from the hyperspectral images were then arranged in a *X* matrix, where the rows and columns of the matrix represent the number of samples, and the number of spectral variables, respectively. The spectra data in the *X* matrix were plotted against the wavelength and visually evaluated to determine the effective spectral region. A bad signal-to-noise ratio (S/N) was observed for a wavelength below 940 nm and above 1670 nm, probably because of the sensitivity of the detector and comparatively higher relative reflectance values over 1670 nm; therefore, the spectral region of 940–1670 nm was used for further data analysis purposes. The average data of each sample in the *X* matrix was preprocessed by using the S–G second derivative preprocessing method to mitigate the baseline shift and improve resolution for a better visualization of spectral differences among the samples with different MCs [26].

Since spectral data are usually represented by a large number of variables (wavebands) which are highly correlated with neighboring variables, a multivariate data analysis method requires to test the relationship between spectral data and the corresponding reference values. Therefore, partial least square regression (PLSR) was used in this study to establish a quantitative model between spectral data of radish slab samples and actual values of MC. The PLSR analysis is particularly suitable when the data set (*X* matrix) contains more spectral variables than the number of observations (samples) [27]. The goal of PLSR is to figure out the linear relationship between the *X* matrix (independent variable) and *Y* variables (reference values, also called dependent variable) using a regression coefficient and error matrix *E*. The PLS equations for *X* and *Y* matrices are described as Equations (6) and (7), respectively.
X = TP^T^ + E(6)
Y = UQ^T^ + F(7)

In this study, PLSR model was constructed by arranging mean spectral data of each sample in a *X* matrix. The corresponding MC values were also arranged in a *Y* matrix. The spectral data matrix *X* was decomposed into the score matrix *T* and the loading matrix *P*, with matrix *E* being the error term. In the same way, the reference value matrix *Y* was decomposed into the score matrix *U* and the loading matrix *Q* with an *F* error term.

The entire set of data (*X* and *Y* matrices) was divided into calibration and validation set. Furthermore, the PLSR model was developed with the calibration set comprised of samples dried by HAD and MD dried samples, and the developed mode was then tested with the validation set comprised of HMCD dried samples. A summary of the descriptive statistics for the calibration and validation data sets is given in Table 2. To prevent the model from underfitting or overfitting, the number of latent vector or factors were selected based on the lowest value of the root mean square error (RMSE) during the cross-validation process by applying the following Equation (8):(8)$$RMSE=\sqrt{\frac{1}{z}\overset{z}{\underset{i=1}{\sum }}{\left(y_{i}-{\hat{y}}_{i}\right)}^{2}}$$
where $y_{i}$ is the actual reference value, ${\hat{y}}_{i}$ is the predicted value from the PLSR, and *z* is the number of predictions.

Furthermore, the performance of the PLSR model was evaluated by calculating the coefficient of determination (*R*^2^) value and root mean square error value for both the calibration and validation sets. In multivariate analysis, outlier identification and removal are important to improve the model accuracy because even a single outlier can have a significant effect on the model compared to the normal sample, providing misleading statistics [28]. Therefore, in this study, outliers were identified through the comparison of concentration residuals (Y_residuals_ = Y_predicted_ − Y_measured_) with the standard error of calibration (SEC) [29,30]. For samples with a residual difference greater than twice the SEC, the samples were considered outliers and eliminated from the data set [29]. All of the chemo metric analyses and computations were executed using MATLAB software version 7.0.4 (The Mathworks, Natick, MA, USA).

### 2.10. Optimal Variable Selection

By taking into consideration the redundancy and multicollinearity of hyperspectral data, it is necessary and important to decrease the high dimensionality of spectral information [14]. The purpose of an optimal variable (wavelength) selection is to extract wavebands composed of important information (in this study, the information is closely related to the MC of the radish slabs) while eliminating the unwanted wavebands from the spectral data, thus reducing the computation time and improving the model performance. In this study, a model-based variable selection method, named variable importance in projection (VIP) was applied for the selection of wavebands that were sensitive to change in the MC of samples. The VIP values based on the PLS weight reflect the important information about the variables contributing to the distinction of the dependent variables from the independent variables [31,32]. The detailed description of the VIP procedure can be found in a previous study done by Akarachantachote et al. [33]. The VIP algorithm was written in MATLAB and executed on the PLSR based results. After performing the VIP analysis, the PLSR model was further developed with only VIP-selected variables.

### 2.11. Visualization of Moisture Distribution

The ultimate advantage of hyperspectral imaging over conventional spectroscopic techniques is to provide spatial information regarding chemical distribution in samples. The generated chemical images can further be used for spatial visualization of the physicochemical properties of the samples. Therefore, in order to create the moisture distribution map for radish slab samples dried by different drying methods, PLSR images for each sample were generated by multiplying the beta (regression) coefficient from the PLSR model developed with VIP-selected variables. For this purpose, the background removal 3D data were reshaped to a 2D matrix, and PLSR images of moisture were generated by using Equation (9). The generated PLS images were processed with a 3 × 3 median filter to enhance the visual display.
(9)$$PLS_{image}=\overset{n}{\underset{z=1}{\sum }}\beta i\times Hi+Constant$$
where *β* is the regression coefficient from the PLSR model developed with VIP-selected variables, *H* is the reshaped 2D data.

## 3. Results and Discussion

### 3.1. Drying Characteristics of Radish Slabs

The moisture ratio curves versus the drying time for radish slabs dried by different drying methods (HAD, MD, and HMCD) are represented in Figure 4. The total drying time required to reach a constant MC was obviously different depending on the drying method. The drying time required to reach a certain MC of approximately 10% (MC_w.b._) was 180, 70, and 45 min in HAD, MD, and HMCD, respectively. Since hot air first dehydrated the surface of the radish slabs, quite a long time was required to remove the internal moisture in radish slabs. As expected, the microwave irradiation as a controlling condition of both MD and HMCD led to a reduction in the total drying time. When microwave and hot air were simultaneously applied to the drying, increasing vapor pressure inside radish slabs by microwave penetration could cause the movement of the internal water to the surface, and eventually hot air could dehydrate the surface.

The moisture ratio data obtained from the drying experiments were applied to seven thin layer drying models to describe the drying behavior of radish slabs under different drying methods. Regardless of drying method, the *R*^2^ and χ^2^ values for seven models were all above 0.95 and below 0.006, respectively. All selected drying models could be appropriate to describe experimental drying data for radish slabs. Among the drying models, the highest R^2^ values (from 0.996 to 0.998), the lowest RMSE values (from 0.014 to 0.021), and χ^2^ values (around 0) were obtained from the Page model for all drying methods. Hence, in spite of similar indices being obtained with other models, the Page model was selected as the most suitable model for describing drying characteristics of radish slabs under different drying methods. However, the simple form of drying model would be convenient for description of the drying characteristics of radish slabs under different drying methods. The calculated coefficients and model indices (R^2^, RMSE, χ^2^, *k*, and *n*) of the Page model for radish slabs are listed in Table 3.

### 3.2. Dielectric Properties of Radish

The DP (dielectric constant ($\varepsilon^{\prime }$) and dielectric loss factor ($\varepsilon^{\prime \prime }$)) of radish slice samples depending on MC at 2450 MHz are summarized in Table 4. The $\varepsilon^{\prime }$ and $\varepsilon^{\prime \prime }$ values of the radish slice sample with a 93.34% MC_w.b._ were 71.82 ± 0.67 and 17.25 ± 1.00, respectively. The $\varepsilon^{\prime }$ and $\varepsilon^{\prime \prime }$ values of radish with a 96% MC_w.b_ measured by Nelson et al. [34] were 67 and 15, respectively, at 2450 MHz. Sipahioglu and Barringer [35] also reported that the $\varepsilon^{\prime }$ and $\varepsilon^{\prime \prime }$ values of radish (95.82% MC_w.b._, temperature range between 20 and 30 °C) at 2450 MHz were 74 to 72 and 17 to 16, respectively. The initial DP of radish slice samples measured in this study was similar to the DP values reported in previous studies; however, the slight differences observed were due to the difference in the MC of the sample used and instrument specifications.

The $\varepsilon^{\prime }$ value of the radish slice sample was decreased with a decreasing MC. This finding clearly showed that the $\varepsilon^{\prime }$ value was heavily dependent on MC. In general, $\varepsilon^{\prime }$ indicates the ability of the material to store the energy in the applied electric field. Therefore, as the MC of the radish slice decreased, microwave energy could pass through the radish slice without being absorbed. The $\varepsilon^{\prime \prime }$ values of the radish sample were stable until around a 60% MC_w.b_ and then they decreased slightly with a decreasing MC. The $\varepsilon^{\prime \prime }$ is closely associated with the ability of the material to dissipate the energy of the applied electric field. The concentrated microwave energy was absorbed inside the radish slices, resulting in the movement of internal moisture in radish slice with increasing internal vapor pressure. According to a previous study [25], when DP of red delicious apple with different MCs were measured at 22 °C, the $\varepsilon^{\prime }$ value was decreased with a decrease in MC; however, the $\varepsilon^{\prime \prime }$ value was stable in the MC range of 79.6% to 55.1% (MC_w.b._).

Based on the measured DP of the radish slice with different MCs, the calculated penetration depth of microwave for radish at 2450 MHz was in the 0.81 and 1.15 cm range.

### 3.3. Overview of the Spectra from Radish Slabs

Figure 5 shows the raw mean spectra of the radish slabs depending on the MC in the wavelength range between 960 and 1680 nm. The spectral reflectance curves of radish slab samples with different MCs showed similar trends through the entire wavelength range tested. As shown in Figure 5a, the intensity of the absorbed relative reflectance gradually decreased with an increase in the MC of radish slabs. Three main absorption regions were observed around 980, 1180, and 1450 nm. The spectral reflectance curve of radish slabs with high MCs (70% to 90% MC_w.b._) was similar to the pure water spectra. Therefore, the Savitzky–Golay (SG) 2nd derivative spectral preprocessing was first applied to raw spectral data. The mean spectra for different MC are represented in Figure 5b. The SG 2nd derivative preprocessing mitigated the baseline effect from the spectra and showed some strong absorption bands at wavelengths of 970, 1160, and 1430 nm. The absorption peak around 970 nm is related to the second overtone of the O-H stretching mode, which represents the MC of the radish slab samples [36]. The peaks around 1140 and 1430 nm represent the combination of the first overtone of the O–H stretching band, which is also related to the MC of the radish sample [37]. The original bands indicating the MC of the radish samples were located around 970, 1180, and 1450 nm, and the band shift was clearly observed when either the SG 1st or 2nd derivative was employed for spectral preprocessing.

### 3.4. PLSR Models

The multivariate analytical model of PLSR was developed with the average spectral data of dried radish slabs for prediction of the MC. For this purpose, the average spectral data with their corresponding MC (reference values) were divided into calibration and validation sets. A summary of the descriptive statistics for the calibration and validation sets is given in Table 2. These statistic values include number of samples, range, mean, and standard deviation (SD). In this study, a total of 112 sample were divided into the calibration and validation sets (76 and 36 samples, respectively).

The PLSR model was initially developed with the calibration set. As mentioned earlier, outliers must be identified and removed during the calibration model development. Thus, a total of four outliers were detected and removed according to the methodology provided in Section 2.9. Thus, the calibration model was redeveloped without using the radish samples detected as outliers, and the efficiency of the calibration model was then tested with a validation set. The optimal number of factors (latent variables) was selected based on the lowest value of predicted root mean square error (RMSE) by the leave-one-out cross-validation process. The results showed that only four PLSR factors could attain a R^2^ value of 0.956 with a RMSE of 4.73% for the calibration set. The predicted values for the MC of the sample are given in Figure 6a. Moreover, the PLSR results from the independent validation data set showed a strong relationship (R^2^ = 0.949) between the measured and the predicted MC values of radish slabs.

In addition to the prediction accuracy, the regression coefficient of the PLSR model, which represents the spectral difference between various groups of samples, can provide the wavelengths that most contribute to the accurate prediction of the desired physicochemical properties of the samples. The highest absolute value of the regression coefficient is considered to be the most important variable responsible for the prediction and interpretation of the model [38]. By visual comparison of the beta coefficient (Figure 6b) with SG 2nd derivative preprocessed spectra (Figure 5b), most of the peaks and the valleys were observed because of the differences in the MC of the dried samples at different times.

By visual comparison of the beta coefficient (Figure 6b) with the SG 2nd derivative preprocessed spectra (Figure 5b), it is possible to observe that most of the peaks and valleys of beta coefficient are due to the differences in MC of the radish slab samples dried for different times.

### 3.5. Optimal Variable Selection for Model Optimization

In this study, the variable importance in projection (VIP)-based variable selection method was used to select the optimal variables and eliminate data redundancy. The PLSR-VIP model was employed to explore the optimal variables that were important for predicting the MC value of radish slab samples. The VIP algorithm was executed on PLSR-based results. Figure 7b shows the importance of each variable for the prediction of the MC of radish slab samples. The higher the VIP score intensity, the more important the variable is. The performance of the PLSR-VIP method may depend on the cutoff value; values greater than one (greater than one rule) are typically used to select the relevant variables. Therefore, in this study, a VIP score greater than one was considered important and the PLSR model was further developed with only 38 VIP-selected variables. The obtained results for both the calibration and validation data sets are depicted in Figure 7a. It was possible to obtain a higher prediction accuracy and a lower prediction error for both the calibration and validation sets using the PLSR-VIP model than using the PLSR model developed with whole variables (124 wavebands) because the VIP-selected variables were mostly obtained from the spectral regions related to the MC of the radish slab samples. The major peaks associated with the MC of the radish slab samples in the VIP plot were similar to the results of a previous study [13]; however, the small shift in the peak location was observed because of the differences in the physical properties of the samples used.

### 3.6. Visualization of Moisture Distribution

The beta coefficient of the PLSR-VIP model was used to build the visual maps of the MC of the radish slab samples dried for different drying times. Figure 8 shows the moisture distribution map of radish slab samples dried by HAD, MD, and HMCD processes over drying time. Different colors correspond to different level of MC in the samples, which is proportional to the spectral differences of the individual pixels. The different MC distribution patterns of the radish samples were observed depending on drying methods. 

At the initial stage of HAD, MC at the edge of the radish slab was lower than in the center; however, the opposite drying pattern was observed in MD. Since hot air dried the samples from the edges to the center, there was a difference between the peripheral and central MC of the samples. On the other hand, MD, which dried the radish slab from the center to the edges, could reduce the differences in the MC at the edges and at the center of the sample. A previous study also reported that no significant difference was observed between the MC in the periphery and at the center of the square shaped mango sample dried by the microwave-vacuum drying method until MC reached 74% [39]. At the initial stage of HMCD, MC in the periphery of the radish slab was slightly lower than in the center; however, the MC in radish samples seemed to be uniformly distributed.

As HAD and MD progressed, non-uniform drying of radish slab samples was clearly observed. For HAD, the edges of the radish slab samples were dried rapidly after a drying time of 30 min, and an excessive shrinkage occurred at the edges in the final stage of drying. The edges of the radish slab were completely dehydrated; however, there was still a high MC in the center. On the contrary, in the case of MD, the drying of the radish slab was relatively uniform until an MC of around 50% was reached. After a drying time of 30 min, the center of the radish slab was dried faster than the edges. In addition, the upper edge of the radish slab was bent at the drying time of 40 min; however, the bending was slightly smoothed out in the final stage of drying. The penetration depth of microwaves in radish was in the 0.81–1.15 cm range which was higher than the thickness of radish slab. Therefore, the excessive heat generated by volumetric heating was accumulated at the center of the radish slab and caused the movement of moisture from the center to the edges as the MD process progressed [13]. The effect of combination drying on moisture distribution can be seen in Figure 8c. A nearly uniform moisture distribution, more uniform than using an individual drying method, was observed. Representatively, a radish slab (20.9%) dried for 40 min showed the nearly uniform drying with little MC difference between the periphery and the center. Since a low MC of the radish slab could result in a low absorption of microwave energy, the effect of HAD was more dominant than the one of MD in the final stage of HMCD. Therefore, it will be necessary to finely tune hot-air temperature and microwave power for obtaining the most uniform drying using HMCD.

The results from this study were similar to the findings of a previous study done by Pu and Sun [13,39]. The loss of MC of hot-air dried mango samples was higher at the edges than in the center, while opposite results were obtained in microwave-vacuum drying. The nearly uniform drying of the mango samples was observed in hot-air and microwave-vacuum combination drying. However, when three different hot-air temperatures were applied to dry potato slices with one fixed diameter and different thickness, the uniform moisture distribution in the slices was obtained from the moisture mapping results regardless of hot-air temperatures and sample conditions [12]. Moreover, the MC in the center of the beef sample was higher than at the edges in the final stage of MD [14]. The drying patterns of potato and beef samples under HD and MD were different from the drying pattern obtained in this study. The drying effect can vary depending on a variety of factors such as the size, shape, and chemical constituents of the samples, and the drying conditions (microwave strength, hot-air temperature, and velocity) [39,40]. Therefore, when different drying techniques are combined to obtain a uniform drying of agricultural products, it is essential to individually evaluate the drying effect of different drying techniques on agricultural products.

## 4. Conclusions

In this study, drying kinetics and the moisture distribution map of radish slab samples under three different drying processes (HAD, MD, and HMCD) were investigated and visualized by HSI technique, respectively. Page model was the most suitable for describing experimental drying data in the all drying processes used, resulting in the highest *R*^2^ values, and in the lowest RMSE and χ^2^ values. Dielectric constant ($\varepsilon^{\prime }$) and dielectric loss factor ($\varepsilon^{\prime \prime }$) of radish at 2450 MHz decreased with decreasing MC. Based on the measured DP, the calculated penetration depth of microwaves for radish was higher than 0.81 cm. The developed HSI system with a spectral rage between 960 and 1680 nm was used to predict and visualize the MC of radish slabs under different drying processes. The quantitative PLSR model with SG 2nd derivative preprocessed technique was established using full wavelength range and yielded good prediction accuracy ($R_{val}^{2}$ of 0.949) and low prediction error (*RMSEV* of 4.92%). Furthermore, when the PLSR-VIP model was developed with only VIP selected variables representing the moisture related variations among the samples, the prediction accuracy for MC in radish slabs was improved. By applying beta coefficients obtained from the PLSR-VIP model, the moisture prediction maps were generated to visualize the level of MC in each pixel of the image. At the final stage of the drying process, HAD and MD led to non-uniform drying of radish slabs with high MC difference between the center and edges. The simultaneous application of hot-air and microwave drying can result in the improvement of drying uniformity during the drying process of radish slabs.

## Figures and Tables

**Figure 1 foods-09-00484-f001:**
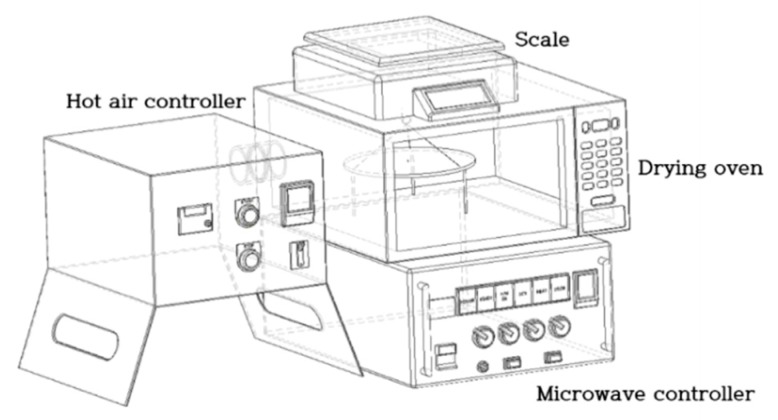
Schematic diagram of microwave and hot-air combination dryer.

**Figure 2 foods-09-00484-f002:**
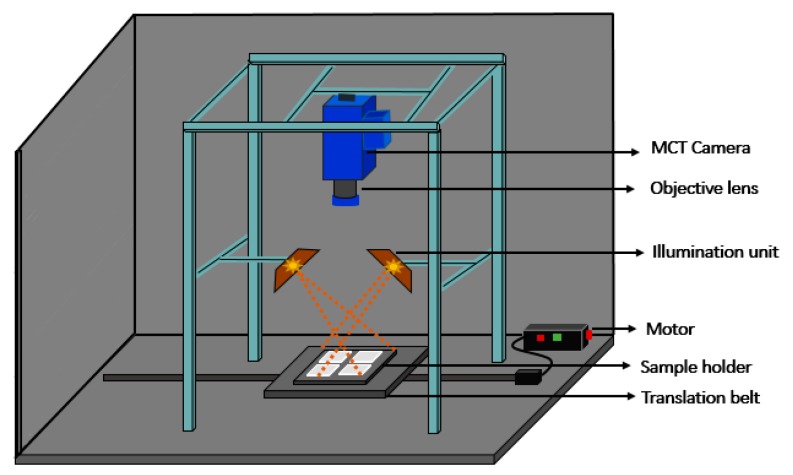
Schematic diagram of the hyperspectral imaging set up for spectral image acquisition of the radish slabs.

**Figure 3 foods-09-00484-f003:**
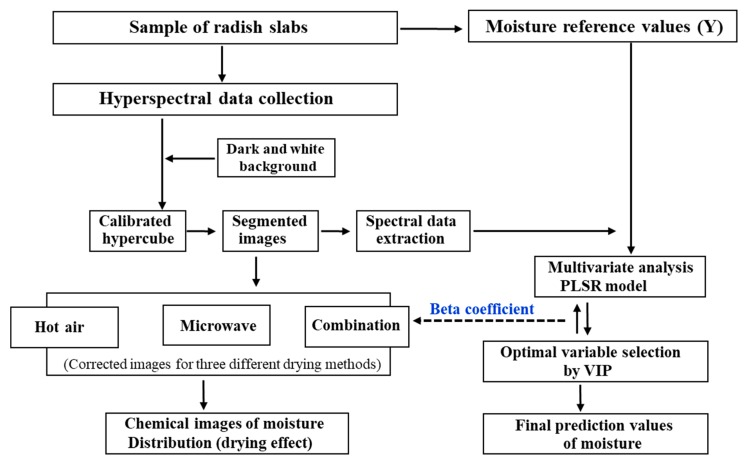
Schematic diagram illustrating the workflow of the data processing.

**Figure 4 foods-09-00484-f004:**
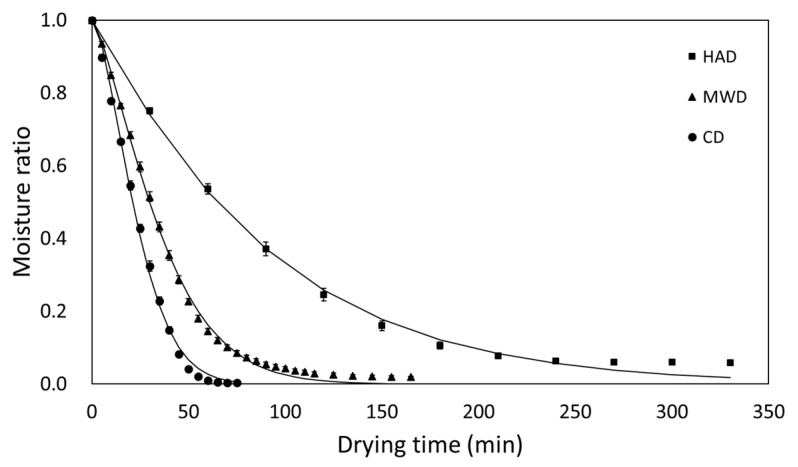
Drying curves of radish slabs undergoing different drying methods (hot-air drying (HAD), microwave drying (MD), and hot-air and microwave combination drying (HMCD)).

**Figure 5 foods-09-00484-f005:**
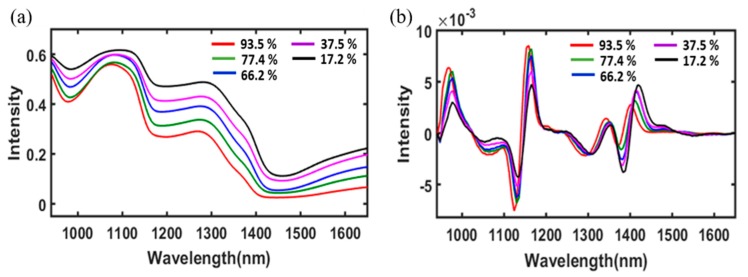
Raw mean spectra (**a**) and Savitzky–Golay (SG) 2nd derivative preprocessed mean spectra (940–1680 nm) of radish slabs with different moisture contents (**b**).

**Figure 6 foods-09-00484-f006:**
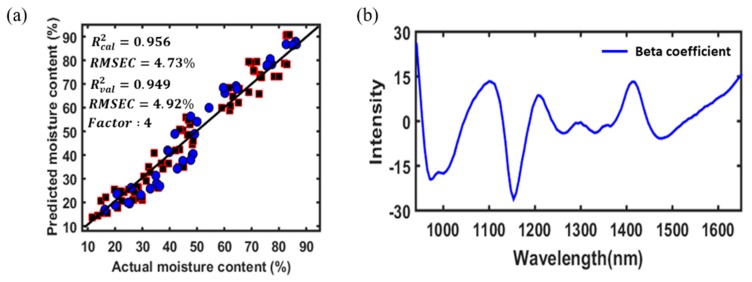
Regression plot from the partial least square regression (PLSR) model for calibration (squares) and validation (circles) sets developed with a spectral range between 960 and 1680 nm (**a**), and the regression coefficient of the PLSR model (**b**).

**Figure 7 foods-09-00484-f007:**
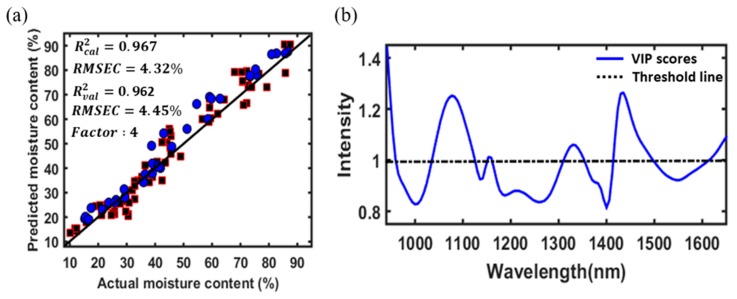
Regression plot using the PLSR model for the calibration (squares) and validation (circles) sets developed with variable importance in projection (VIP) selected variables (**a**), and VIP score plot calculated determined using the PLSR model (**b**).

**Figure 8 foods-09-00484-f008:**
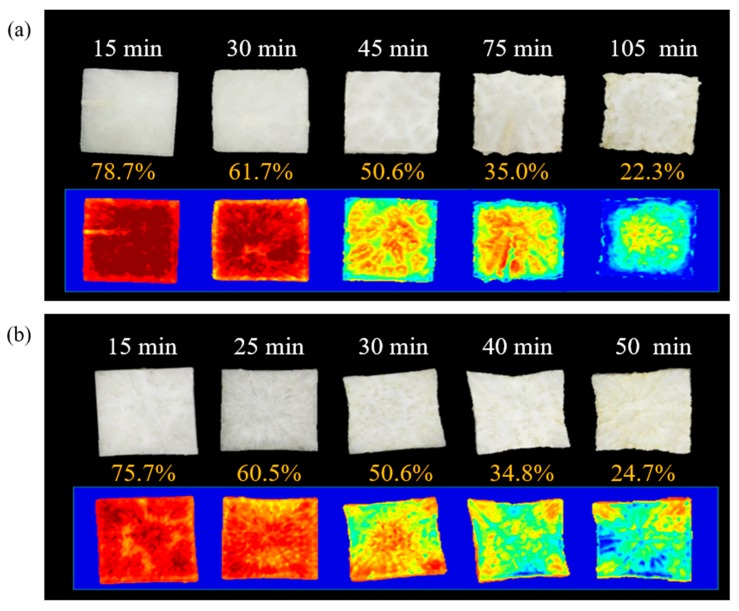
Moisture distribution map of radish slabs during different drying methods: (**a**) hot-air (HAD), (**b**) microwave (MD), (**c**) combination drying (HMCD).

**Table 1 foods-09-00484-t001:** Drying models used for describing the drying characteristics of radish slabs. MR: moisture ratio.

Model Name	Equation	References
Lewis	MR =exp(−kt)	[17]
Page	$\mathrm{MR}\;=\exp\left(-kt^{n}\right)$	[18]
Henderson and Pabis	MR =aexp(−kt)	[19]
Logarithmic	MR =aexp(−kt)+c	[20]
Two-term	$\mathrm{MR}\;=a\exp\left(-kt\right)+b\exp\left(-k_{0}t\right)$	[21]
Midilli et al.	MR =aexp(−kt)+bt	[22]
Wang and Singh	$\mathrm{MR}\;=1+at+b\left(t^{2}\right)$	[23]

**Table 2 foods-09-00484-t002:** Summary of descriptive statistics for the calibration and validation data sets (excluding number of samples, all units in %).

	No. of Samples	Minimum	Maximum	Mean ± SD
Calibration set	76	13.6	90.6	46.8 ± 23.4
Validation set	36	17.2	86.5	49.9 ± 23.6

Std: standard deviation; in the calibration set, four samples were identified as outliers and removed.

**Table 3 foods-09-00484-t003:** The calculated coefficients of the applied drying models for radish slabs. RMSE: root mean square error.

Name of Model		Drying Methods		
		Hot Air	Microwave	Combination
Lewis [17]	*k*	0.0112	0.0267	0.0399
	$R^{2}$	0.9945	0.9759	0.9519
	RMSE	0.0233	0.0486	0.0770
	$\chi^{2}$	0.0005	0.0024	0.0059
Page [18]	*k*	0.0075	0.0063	0.0054
	*n*	1.0854	1.3837	1.5872
	$R^{2}$	0.9960	0.9981	0.9970
	RMSE	0.0210	0.0142	0.0199
	$\chi^{2}$	0.0004	0.0002	0.0004
Henderson and Pabis [19]	*a*	1.0203	1.0996	1.1065
	*k*	0.0114	0.0292	0.0437
	$R^{2}$	0.9950	0.9847	0.9634
	RMSE	0.0233	0.0394	0.0695
	$\chi^{2}$	0.0005	0.0016	0.0048
Logarithmic [20]	*a*	1.0131	1.1171	1.2566
	*k*	0.0118	0.0269	0.0294
	*c*	−0.4470	−0.0306	−0.1954
	$R^{2}$	0.9953	0.9867	0.9863
	RMSE	0.0239	0.0374	0.0441
	$\chi^{2}$	0.0006	0.0014	0.0019
Two-term [21]	*a*	13.7366	1.0657	35.0077
	*k*	0.0102	0.0292	0.0184
	*b*	−12.7175	0.0339	−33.9479
	$k_{0}$	0.0101	0.0292	0.0179
	$R^{2}$	0.9951	0.9847	0.9881
	RMSE	0.0259	0.0410	0.0428
	$\chi^{2}$	0.0007	0.0017	0.0018
Midili et al. [22]	*a*	1.0252	1.091	1.0647
	*k*	0.0117	0.028	0.0342
	*b*	0.0000	0.000	−0.0019
	$R^{2}$	0.9955	0.986	0.9843
	RMSE	0.0233	0.039	0.0472
	$\chi^{2}$	0.0005	0.001	0.0022
Wang and Singh [23]	*a*	−0.0079	−0.0177	−0.0287
	*b*	0.000	0.0001	0.0002
	$R^{2}$	0.9861	0.9739	0.9934
	RMSE	0.0389	0.0515	0.0294
	$\chi^{2}$	0.0015	0.0027	0.0009

**Table 4 foods-09-00484-t004:** Dielectric properties of radish depending on moisture content at 2450 MHz.

Moisture Content (%, MC_w.b._)	Dielectric Constant ($\varepsilon^{\prime }$)	Dielectric Loss Factor ($\varepsilon^{\prime \prime }$)	Penetration Depth (cm)
93.34 (± 0.65)	71.82 (± 0.67)	17.25 (± 1.00)	0.97 (± 0.06)
90.33 (± 0.82)	68.63 (± 2.51)	16.29 (± 1.46)	1.00 (± 0.10)
80.85 (± 2.75)	64.72 (± 2.08)	17.19 (± 0.91)	0.92 (± 0.06)
69.76 (± 0.94)	62.49 (± 1.96)	19.39 (± 1.81)	0.81 (± 0.08)
59.95 (± 0.96)	58.10 (± 3.53)	17.59 (± 1.76)	0.86 (± 0.11)
51.08 (± 1.52)	53.62 (± 4.87)	14.57 (± 0.23)	0.99 (± 0.03)
41.82 (± 0.74)	46.49 (± 3.49)	15.99 (± 0.74)	0.84 (± 0.05)
30.78 (± 0.34)	36.06 (± 3.49)	10.63 (± 2.61)	1.14 (± 0.19)
18.75 (± 1.52)	24.88 (± 4.21)	8.59 (± 0.57)	1.15 (± 0.08)

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
