# Peer review of "Determination of Drying Patterns of Radish Slabs under Different Drying Methods Using Hyperspectral Imaging Coupled with Multivariate Analysis"

_foods, 2020, doi:10.3390/foods9040484_

Round 1

Reviewer 1 Report

This manuscript is very interesting and I think it is well organized. The visualization of the drying patterns by different methods is expected to find the ideal drying method for foods.

Author Response

This manuscript is very interesting and I think it is well organized. The visualization of the drying patterns by different methods is expected to find the ideal drying method for foods.

Answer: The authors appreciate the reviewer’s constructive comments and suggestions. 

Reviewer 2 Report

The manuscript titled "Determination of drying patterns of radish slabs under different drying methods using hyperspectral imaging coupled with multivariate analysis", presents the drying characteristics of radish slabs under different drying methods and problems of microwave and hot-air drying. The title of the manuscript is consistent with the content of work. The study was well conducted and has some interesting results which can be of interest readers. The downside of this paper is that it does not discuss qualitative aspects and nutritional value in raw and dried radish after microwave and hot-air drying. In the current state, I recommend to reconsider after major revision the manuscript, because it fails to point out novel insights and does not add much to the knowledge base. The authors provide a rather technical report of an experiment without focusing on a clear research question. I have several suggestions to improve the quality of the manuscript detailed below.

The abstract provides too much details on experimental procedures and results, but misses to frame a clear research question and to focus on highlighting key results and insights. The introduction is too unfocused and does not lead to a research question. Moreover, a focused discussion of the obtained results in the frame of a research question is missing. Conclusion is essentially a summary of the results. A true conclusion with respect to a clear research question is missing. Formal way of writing should be used, so forms like "carried out" should be avoided. English should also be carefully checked and corrected, preferably by a native speaker.

Author Response

The manuscript titled "Determination of drying patterns of radish slabs under different drying methods using hyperspectral imaging coupled with multivariate analysis", presents the drying characteristics of radish slabs under different drying methods and problems of microwave and hot-air drying. The title of the manuscript is consistent with the content of work. The study was well conducted and has some interesting results which can be of interest readers. The downside of this paper is that it does not discuss qualitative aspects and nutritional value in raw and dried radish after microwave and hot-air drying. In the current state, I recommend to reconsider after major revision the manuscript, because it fails to point out novel insights and does not add much to the knowledge base. The authors provide a rather technical report of an experiment without focusing on a clear research question. I have several suggestions to improve the quality of the manuscript detailed below. 

Answer: The authors appreciate the reviewer's constructive comments and suggestions. The manuscript was revised to incorporate the review comments. In this study, we focused on the determination of moisture distribution of radish slabs during different drying processes. For future study, we will determine the change in nutritional value of radish using HSI technique. 

The abstract provides too much details on experimental procedures and results, but misses to frame a clear research question and to focus on highlighting key results and insights. The introduction is too unfocused and does not lead to a research question. Moreover, a focused discussion of the obtained results in the frame of a research question is missing. Conclusion is essentially a summary of the results. A true conclusion with respect to a clear research question is missing. Formal way of writing should be used, so forms like "carried out" should be avoided. English should also be carefully checked and corrected, preferably by a native speaker.

Answer: The abstract was revised to incorporate the review comments. The introduction part was revised to explain why this study was conducted.  The discussion and conclusion parts were revised to focus on the highlighting results and explaining MC distribution in radish slabs. The manuscript was checked and corrected by native English speaker, and the certificate of English editing was attached.

Reviewer 3 Report

This manuscript reports the use of NIR and hyperspectral imaging to evaluate the drying processes of radishes by determining the moisture content.  The manuscript is technically sound and will be of interest to researchers using spectroscopy to evaluate foods.  The method described can be adapted to a variety of foods. 

Author Response

This manuscript reports the use of NIR and hyperspectral imaging to evaluate the drying processes of radishes by determining the moisture content.  The manuscript is technically sound and will be of interest to researchers using spectroscopy to evaluate foods.  The method described can be adapted to a variety of foods. 

Answer: The authors appreciate the reviewer’s constructive comments and suggestions.

Reviewer 4 Report

I found only a few minor errors that need to be corrected.

  • The color markers be each keyword are not required
  • Line 97: The latin names of plant should be written in italic font.
  • Could the authors insert the descriptions of each particular drying methods in separate subsections in the "2.3. Drying procedure"

Author Response

I found only a few minor errors that need to be corrected.

Answer: The authors appreciate the reviewer’s constructive comments and suggestions. The manuscript was revised to incorporate the review comments.

  • The color markers be each keyword are not required
  • Answer: The markers were deleted.
  • Line 97: The latin names of plant should be written in italic font.
  • Answer: Latin name of the plant was written in Italic font.
  • Could the authors insert the descriptions of each particular drying methods in separate subsections in the "2.3. Drying procedure"
  • Answer: The descriptions of each drying methods was divided into subsections (2.3.1 HAD procedure, 2.3.2 MD procedure, 2.3.3 HMCD procedure) of “2.3. Drying procedure”
  •  

Round 2

Reviewer 2 Report

The revisions are satisfactory and the revised manuscript can be accepted to publish.